# Virucidal Activity of Lemon Essential Oil against Feline Calicivirus Used as Surrogate for Norovirus

**DOI:** 10.3390/antibiotics12020322

**Published:** 2023-02-03

**Authors:** Francesco Pellegrini, Michele Camero, Cristiana Catella, Giuseppe Fracchiolla, Sabina Sblano, Giovanni Patruno, Claudia Maria Trombetta, Michela Galgano, Annamaria Pratelli, Maria Tempesta, Vito Martella, Gianvito Lanave

**Affiliations:** 1Department of Veterinary Medicine, University of Bari Aldo Moro, Valenzano, 70010 Bari, Italy; 2Department of Pharmacy-Drug Sciences, University of Aldo Moro of Bari, 70125 Bari, Italy; 3Department of Molecular and Developmental Medicine, University of Siena, 53100 Siena, Italy

**Keywords:** feline calicivirus, virucidal activity, lemon essential oil

## Abstract

Norovirus (NoV) is regarded as a common cause of acute gastrointestinal illness worldwide in all age groups, with substantial morbidity across health care and community settings. The lack of in vitro cell culture systems for human NoV has prompted the use of cultivatable caliciviruses (such as feline calicivirus, FCV, or murine NoV) as surrogates for in vitro evaluation of antivirals. Essential oils (EOs) may represent a valid tool to counteract viral infections, particularly as food preservatives. In the present study, the virucidal efficacy of lemon EO (LEO) against FCV was assessed in vitro. The gas chromatography hyphenated with mass spectrometry (GC/MS) technique was used to reveal the chemical composition of LEO. The following small molecules were detected as major components of LEO: limonene (53%), β-pinene (14.5%), γ-terpinene (5.9%), citral (3.8%), α-pinene (2.4%), and β-thujene (1.94%). LEO at 302.0 μg/mL, exceeding the maximum non cytotoxic limit, significantly decreased viral titre of 0.75 log_10_ TCID50/50 μL after 8 h. Moreover, virucidal activity was tested using LEO at 3020.00 μg/mL, determining a reduction of viral titre as high as 1.25 log_10_ TCID50/50 μL after 8 h of time contact. These results open up perspectives for the development of alternative prophylaxis approaches for the control of NoV infection.

## 1. Introduction

The emergence of antiviral drug resistance and the outbreak of the coronavirus disease 2019 (COVID-19) pandemic [1] have prompted the research for novel natural and synthetic compounds with antiviral properties. Among these substances, essential oils (EO) have been included in strategies for the development of new antivirals. EOs are natural products and possess a heterogeneous chemical composition mainly derived from benzene and terpenes [2,3]. EOs are aromatic oily liquids derived from plant material (flowers, buds, seeds, leaves, branches, bark, grass, wood, fruit, and roots) by steam distillation or pressing, fermentation, enfleurage, or extraction by heat or solvents. The antimicrobial, antiviral, antioxidant, anti-inflammatory, and anticancer activities of EOs have been consistently reported [4,5,6,7]. EOs have long been used as preservatives and flavor enhancers of foods and beverages [8], and also in the perfume and cosmetic industries. Furthermore, the virucidal activities of these compounds are also exploited in the food and healthcare sectors, to prevent the transmission of viral pathogens. 

Today, the attention of consumers towards food quality and safety has risen. Despite precise hygiene standards guaranteed by food business operators, foodborne diseases still represent a serious global public health concern. According to the World Health Organization (WHO), 600 million people worldwide become ill after ingesting contaminated food, with 420,000 deaths every year [9].

Moreover, given the growing interest in food preservation, the current trend is to consume healthy foods without artificial preservatives, with EOs representing a valid alternative. The virucidal efficacy of EOs against enveloped RNA and DNA viruses, i.e., Herpes simplex virus types 1 and 2 (HSV-1 and HSV-2), Caprine herpesvirus 1 (CpHV-1), dengue type 2 (DEN-2), Junin virus (JUNV), influenza virus, and coronavirus (CoV) has been reported elsewhere [10,11,12,13]. Controversial data are available on the efficacy of EOs against non-enveloped viruses [14,15]. Many studies have reported that EOs are not able to reduce the viral titer of non-enveloped viruses [16], while the virucidal effects of EOs have been demonstrated against enteroviruses [17], norovirus (NoV) surrogates [18], and rotaviruses [19].

NoVs compose a genus (*Norovirus*) of the *Caliciviridae* family and possess naked virus-like particles with single-stranded RNA genomes. NoVs are regarded as one of the most widespread agents of acute gastroenteritis of viral origin, thus representing the major leading cause of foodborne illnesses [20]. NoVs are highly infectious, with 10–100 viral particles being sufficient to infect an individual [21]. They are resistant in the environment and cause approximately 700 million cases of disease and 200,000 deaths annually worldwide [22]. The clinical signs appear after a short incubation period (10–51 h) and include stomach cramps, diarrhea, and vomiting, which usually last around 2–3 days, although virus shedding can be much longer [20]. The predominant genotype in humans is GII.4, with different pandemic variants emerging over the years, with the latter being GII.4 Sydney (2012) variant [23,24]. 

NoVs are difficult to control and stringent sanitary measures must therefore be applied to prevent and contain the diffusion. NoV infections mainly occur in community settings, i.e., hospitals, nursing homes, schools, and also in confined spaces, such as commercial and cruise ships [25,26]. Transmission directly occurs from person to person, via the fecal-oral route, aerosols, infected food and water, or contact with contaminated surfaces. Foodborne outbreaks are often associated with the consumption of raw seafood, salads, berries, contaminated water, cold foods, sprouts, herbs, and spices [23,27]. In compliance with hygiene measures applied to the handling and distribution of food and drinks, natural, non-toxic, and low environmental impact disinfectants are available to counteract NoV spread on the surfaces of fomites and food.

NoV treatment is mainly based on support therapy. Thus far, no specific anti-NoV drug candidates have passed clinical trials [23,24]. Nitazonxanide, an FDA-approved drug, was proven to decrease the duration of NoV-induced symptoms and has been applied to several NoV patients [23,24]. More interestingly, there are currently many vaccine candidates in development, with the great challenge posed by NoV genetic/antigenic diversity [28,29]. 

Human NoVs are not cultivatable in common cell lines, and, after decades of attempts and failures, a culture system has been developed using enteroid cells of human origin [30]. Yet, few laboratories have trained personnel and equipment to use the enteroid system. Furthermore, the replication of human NoVs is time consuming and labor-intensive. Feline calicivirus (FCV), a member of a distinct *Caliciviridae* genus (*Vesivirus*), is easily cultivatable on feline kidney cells and it is not zoonotic. Accordingly, FCV has been used as the preferred surrogate for NoV since the 1970s [31]. In the mid-2010s, a cultivatable NoV (murine NoV, MNoV), more genetically related to human NoV than FCV, was discovered in mice with impaired functionality of the immune system [28]. Due to the difficulties of cultivating human NoVs, virus surrogates have been used as a good proxy to assess the virucidal activity of different substances against NoVs [32,33]. In several studies, the anti-viral activity of EOs in vitro has been investigated using either FCV or MNoV [18,34,35].

Different EOs, including lemon EO (LEO), added to fruit berry packages, have been shown to reduce the viral loads of hepatitis A virus, a non-enveloped viral pathogen, after 1 h of incubation at room temperature [36]. Similarly, the citrate contained in lemon juice and used as a component of disinfectants has been recently reported to affect the morphology of NoV-like particles [37].

The aim of the present study was to assess the virucidal efficacy of LEO against FCV in vitro.

## 2. Materials and Methods

### 2.1. Analysis of LEO

The pure *Citrus lemon* essential oil, also named as LEO, extracted from lemon peel, was provided by Specchiasol S.r.l. (Bussolengo, VR, Italy) and stored in a brown glass bottle at a temperature of 0–4 °C. Solvents (in analytical grade), n-alkanes standard mixture C10–C40, and all standard compounds were purchased from Supelco Sigma-Aldrich S.r.l. (Milano, Italy). Filters were supplied by Agilent Technologies Italia S.p.a (Milano, Italy). The composition of commercially available LEO used in our experiments was confirmed by hyphenated gas chromatography with mass spectrometry (GC/MS) technique [38,39].

### 2.2. Gas Chromatography/Mass Spectrophotometry (GC/MS)

Chromatographic analyses of LEO were performed on an Agilent 6890 N gas chromatograph equipped with a 5973 N mass spectrometer, provided with a HP-5 MS (5% phenylmethylpolysiloxane, 30 m, 0.25 mm i.d., 0.1 μm film thickness (J & W Scientific, Folsom) capillary column. The following temperature programmer was used: 5 min at 60 °C, then 4 °C/min to 220 °C, then 11 °C/min to 280 °C, held for 15 min, for a total run of 65 min. Injector and detector temperatures were 280 °C; the carrier gas was He; the flow rate was 1 mL/min; the split ratio was 1:50; the acquisition range was 29–400 *m*/*z* in electron-impact (EI) mode; and the ionization voltage was 70 eV [11].

### 2.3. Compound Identification

For chemical characterization, LEO was diluted 1:100 in ethyl acetate and after filtration, 1 μL of EO solution was injected into the GC-MS. Qualitative analyses were carried out comparing the calculated Linear Retention Indices (LRIs) and Similarity Index of Mass Spectra (SI/MS) for the obtained peaks with the Arithmetic Index (AI) and the analogous data reported in the literature [40] and in the NIST 2017 Databases (NIST 17, 2017. Mass Spectral Library-NIST/EPA/NIH. Gaithersburg, USA: National Institute of Standards and Technology. Last access 12_2022), respectively. The LRI of each compound was determined by temperature programming analysis and was calculated as previously described [41] with an equation related to a homologous series of n-alkanes (C10–C40) under the same operating conditions. SI/MS were determined as previously reported [41,42,43]. Component relative percentages were calculated based on GC peak areas without using correction factors.

### 2.4. Cells and Virus

Crandell Rees Feline Kidney (CrFK) cells were cultured at 37 °C in a 5% CO_2_ atmosphere in Dulbecco-MEM supplemented with 10% fetal bovine serum, 100 IU/mL penicillin, 0.1 mg/mL streptomycin, and 2 mM l-glutamine. The same medium was used for the antiviral assays. The FCV field strain 283/12 was cultured and titrated on CrFK cells. The virus stock with a titer of 10^8^ Tissue Culture Infectious Dose–(TCID_50_)/50 μL was stored at −80 °C and used for the experiments.

### 2.5. Cytotoxicity Assay

The cytotoxicity of LEO was assessed using an in vitro Toxicology Assay Kit (Sigma–Aldrich Srl, Milan, Italy), based on 3-(4,5-dimethylthiazol-2 yl)-2,5-diphenyl tetrazolium bromide (XTT). The assay was previously performed as previously described [44]. Confluent 24-h monolayers of CrFK cells grown in 96-well plates were used to assess the cytotoxicity of LEO at different concentrations (8360, 4180, 2090, 1045, 522.50, 261.25, 130.63, 65.31, 32.66 μg/mL). In all experiments, untreated cells and cells treated with equivalent dilutions of DMSO without LEO were used as the control and the vehicle control, respectively. The percentage of cytotoxicity was calculated using the formula: % Cytotoxicity = [(OD of control cells−OD of treated cells) ×100]/OD of control cells.

The maximum non-cytotoxic concentration was assessed and regarded as the concentration at which viability of the treated CrFK cells decreased to 20% with respect to the control cells (CC_20_). The experiments were performed in triplicate.

### 2.6. Virucidal Activity Assay

The potential inhibitory effect of LEO against FCV was evaluated by pre-treatment of the virus with LEO at a maximum non-cytotoxic dose of 30.20 μg/mL and over the cytotoxic threshold (302.00 and 3020.00 μg/mL). In detail, 100 μL of FCV at stock titer were treated with LEO (1 mL) at different concentrations at room temperature. After 10 min, 30 min, 1 h, 4 h, and 8 h, the different mixtures of virus-LEO and untreated infected cells (control virus) were subjected to viral titration in CrFK cells.

### 2.7. Viral Titration

Ten-fold dilutions (up to 10^−8^) of each supernatant were titrated in quadruplicates in 96-well plates containing CrFK cells. The plates were incubated for 72 h at 37 °C in a 5% CO_2_ environment. Based on cytopathic effect, viral titer was calculated.

### 2.8. Data Analysis

LEO concentrations were converted in log_10_ and cytotoxicity assay results were evaluated by a non-linear curve fitting. Moreover, a dose-response curve was elaborated through non-linear regression analysis in order to evaluate goodness of fit.

From the fitted dose response curves achieved in each experiment, CC_20_ was assessed. Data obtained from GC/MS and virucidal assays were reported as area % ± SEM and mean ± SD, respectively.

Normality of distribution was evaluated by Shapiro-Wilk test. Data were analyzed by T-Student test for independent samples or One way Analysis of Variance (ANOVA), followed by a Bonferroni test as a post hoc test (statistical significance set at 0.05). Statistical analyses were carried out by GraphPad Prism v8.1.2 program Intuitive Software for Science, San Diego, CA, USA. 

## 3. Results

### 3.1. Analytical Details of LEO

The analysis of LEO revealed a complex mixture mainly consisting of oxygenated and hydrocarbon monoterpenes. There were 21 different components accounting for 87.81% of the mixture. The six major detected compounds were limonene (53%), β-pinene (14.5%), γ-terpinene (5.9%), citral (3.8%), α-pinene (2.4%), and β-thujene (1.94%). The composition of the LEO has been summarized in Table 1.

### 3.2. Cytotoxicity

The cytotoxicity of LEO was determined by microscopic examination of cell morphology and measurement of cell viability by the XTT colorimetric method after exposing the cells to various concentrations of the compound (8360, 4180, 2090, 1045, 522.50, 261.25, 130.63, 65.31, 32.66 μg/mL) for 72 h. The intensity and variety of the cellular morphological changes (loss of cell monolayer, granulation, cytoplasmic vacuolization, stretching and narrowing of cell extensions, and darkening of the cell borders) were dose dependent [44]. Cytotoxicity was assessed by spectrophotometrically measuring the absorbance signal. In all of the experiments, DMSO did not show any effect on cells. Based on fitted dose–response curves, the CC_20_ value of LEO was assessed at 30.20 μg/mL.

### 3.3. Virucidal Activity

The virucidal effects of LEO at different concentrations (30.20, 302.00, and 3020.00 μg/mL) and contact times (10 min, 30 min, 1 h, 4 h, and 8 h) at room temperature were evaluated against FCV and compared with the control virus. LEO at 30.20 μg/mL showed non-significant reductions in viral titre (0–0.5 log_10_TCID_50_/50 μL, *p* > 0.05) at all of the time contacts (Figure 1A) as compared to the virus control (6.75–7.00 log_10_ TCID_50_/50 μL). After 8 h, LEO at 302.00 μg/mL determined a significant decrease in viral titre of 0.75 log_10_TCID50/50 μL (*p* < 0.05) with respect to the virus control (6.75–7.00 log_10_ TCID_50_/50 μL) (Figure 1B). LEO at 3020.0 μg/mL induced significant reductions in viral titre of 1.00 log_10_TCID_50_/50 μL (*p* < 0.05) at 4 h and of 1.25 log_10_TCID_50_/50 μL (*p* < 0.05) at 8 h when compared to the virus control (6.75 log_10_ TCID_50_/50 μL) (Figure 1C). In the ANOVA model, the results of viral titration of FCV treated with LEO at different concentrations (30.20, 302.00, and 3020.00 μg/mL) were compared with the virus control, showing a statistically significant effect (F = 15.13, *p* < 0.05).

## 4. Discussion

Natural products and their structural analogs have historically provided a contribution to pharmacotherapy, particularly for the study of infectious diseases. Natural products pose challenges for drug discovery due to difficulties in screening, isolation, characterization, and optimization that contributed to a decline in their development by the pharmaceutical industry from the 1990s onwards [45]. In recent years, numerous technological and scientific achievements have dealt with such difficulties and paved the way to new perspectives. Accordingly, interest in natural products as drug leads has been renovated. 

Moreover, the COVID-19 pandemic has also prompted research for adequate alternative disinfection procedures. Beyond their well-recognized properties, essential oils could be used as safe natural disinfectant agents [13]. EOs have also been used in the food sector as natural preservatives of salads, fruits, and berries [18,46]. Before being packaged and placed onto the market, these food products are washed, and chemical additives are added, thus increasing the risk of toxic by-products on food surfaces and prompting the research of adequate alternatives. 

The antiviral activity of EOs has been investigated against NoV surrogates [18,35,47], among which is FCV, which has often been used to evaluate the efficacy of food preservatives in the industry [48,49]. 

In this study, we tested the virucidal activity of LEO against FCV in vitro on CrFK cells. For this evaluation, we used LEO at different concentrations, including the maximum non-cytotoxic dose (30.20 μg/mL) and 10- and 100-fold concentrations exceeding the cytotoxic threshold (302.00 and 3020.00 μg/mL, respectively) to assess its potential use as a surface disinfectant. The maximum non-cytotoxic dose must be considered when EOs are added to mouthwashes, personal hygiene products, soaps, perfumes, and cosmetics. In vitro inactivation of FCV occurred in a dose-dependent and time contact fashion. At the maximum non-cytotoxic concentration (30.20 μg/mL), no significant decrement of viral titre occurred. LEO at 302.0 μg/mL significantly decreased viral titre of 0.75 log_10_ TCID_50_/50 μL after 8 h, while LEO at 3020.00 μg/mL significantly reduced viral titre of 1 log_10_ TCID_50_/50 μL after as early as 4 h of contact time, reaching 1.25 log_10_ TCID_50_/50 μL at 8 h.

To date, there has been a high number of studies regarding the antibacterial activity of EOs [46], but only a few studies have been conducted on their virucidal activity [15]. Among these studies, clove, oregano, and zataria EOs were tested in vitro against NoV surrogates. For instance, 2% oregano EO (OEO) for 2 h at 37 °C was able to decrease FCV and MNoV titers by 3.75 log_10_ TCID_50_/mL and 1.04–1.62 log_10_ TCID_50_/mL, respectively [18]. OEO reduced MNoV infectivity by 1.07 log_10_ after 24 h and carvacrol, a single compound in OEO, was able to decrease MNoV by up to 3.87 log_10_ within 1 h. Upon transmission electron microscopy, both compounds were able to disintegrate virus capsid and, subsequently, the RNA [47]. Recently, artemisia EO determined 48% in vitro inhibition on FCV and 64% on MNoV at 0.1 and 0.01%, respectively [34]. Lemongrass EO preincubated with MNoV exhibited a significant reduction in viral plaque formation in a time and dose-dependent manner [35].

The results of this study are difficult to compare with those from other reports due to the different conditions used in the experiments; i.e., temperatures, the number of viral particles, and the EO/virus contact times. The conservation of EOs is also a limiting factor as they are very sensitive to heat, light, oxygen, and humidity, and are characterized by a remarkable rapidity of evaporation. Furthermore, variations in the composition of EOs may depend on the soil where the plant grew, from which the oil is extracted [50,51], and standardization of the chemical composition of EOs is difficult.

Despite the significant virucidal effects of LEO against FCV reported in this study, the results are not as noticeable as those observed for other EOs against enveloped viruses. To date, only a few studies have demonstrated good antiviral efficacy of EOs against non-enveloped viruses, such as coxsackievirus, poliovirus, human NoV, MNoV, astroviruses, and hepatitis viruses [52,53], with the antiviral activity being mainly targeted to the capsid [54,55]. Structural changes in the capsid of FCV have been observed after exposure to cranberry juice and cranberry proanthocyanidins [54]. NoV particles treated with grape seed extract also appeared altered under electron microscope visualization [55]. A similar mechanism of action could be hypothesized for LEO against FCV. In another study in vitro, hyssop and marjoram EOs were not able to reduce or inactivate emerging food-borne non-enveloped viruses [16]. 

EOs have proved more effective against enveloped viruses (i.e., influenza virus, herpes virus, and SARS CoV-2) [56]. The mechanisms of action of EOs against these viruses have not yet been fully elucidated, but it can be hypothesized a virucidal effect due to alterations of the envelope glycoproteins that are necessary for virus adsorption and entry into host cells [53]. Electron microscopy observation has demonstrated the disaggregation of the HSV-1 envelope after pretreatment with Eos [57]. 

FCV, as with other caliciviruses, is highly resistant to environmental conditions and antimicrobials [58,59]. In this study, the untreated FCV used as the control after 8 h at room temperature did not show any relevant decrease in viral titer. NoV is also able to survive on many environmental surfaces for weeks or months at room temperature [60], and its capsid proved highly resistant to lipophilic disinfectants (e.g., quaternary ammonium compounds) and solvents (e.g., alcohol) [61].

The chemical composition of LEO revealed the presence of 21 distinct molecules, the main fractions of which were limonene, β-pinene, γ-terpinene, citral, α-pinene, and β-thujene. EOs are complex mixtures which can be toxic at high concentrations, especially if orally taken [62]. In order to reduce the cytotoxicity of LEO, it would be interesting to identify the active molecules and to individually test them. Limonene accounted for more than 50% of components of the LEO used in this study, and is often identified in EO composition [47,63]. Limonene and β-pinene fractions demonstrated significant virucidal effects against MNoV [64]. Moreover, 2% and 4% citral concentrations significantly decreased (73.09%) MNoV infectivity after 6 and 24 h of exposure [35]. Other less represented fractions of LEO could also be tested to assess their antiviral activity.

In conclusion, we demonstrated the antiviral activity of LEO using a cultivatable NoV surrogate, FCV. LEO was able to significantly decrease FCV infectivity in our experimental design. Further studies are needed to improve the performance of LEO, to investigate the mechanisms of action on FCV, and to assess if other FCV strains may have different susceptibility to LEO. Furthermore, these results open up different perspectives for the development of prophylaxis tools for the disinfection and sanitization of surfaces of fomites, and for decreasing the risks of exposure to NoV in foods.

## Figures and Tables

**Figure 1 antibiotics-12-00322-f001:**
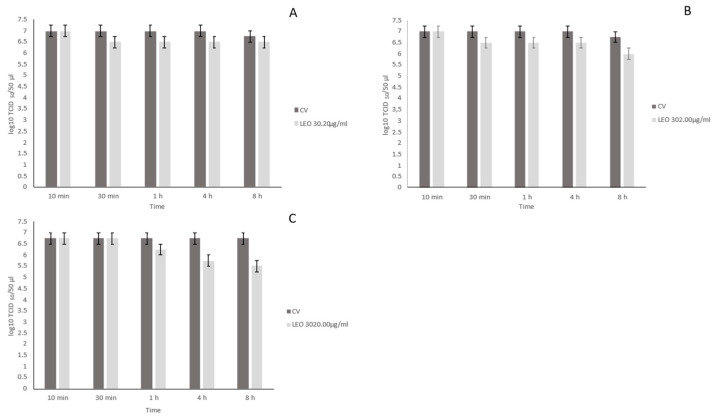
Virucidal effect of lemon essential oil (LEO) incubated with Feline Calicivirus (FCV) for 10 min, 30 min, 1 h, 4 h, and 8 h at room temperature and subsequently titrated in Crandell-Reese Feline Kidney (CrFK) cells. LEO was used at 30.20 μg/mL (**A**), 302.00 μg/mL (**B**), and 3020.00 μg/mL (**C**) against FCV. Viral titres of FCV were expressed as log_10_ TCID_50_/50 μL and plotted against LEO at different concentrations. Bars in the figures indicate the means. Error bars indicate the standard deviation.

**Table 1 antibiotics-12-00322-t001:** Detailed description of EOs chemotype. In bold the predominant components.

N	Components	LRI	AI	*Citrus lemon*	
				Area ± SEM	SI/MS
1	Ethyl propanoate	714	714	0.1 ± 0.010	91
**2**	**α-pinene ^a^**	**930**	**931**	**2.4 ± 0.5**	**95**
**3**	**β-thujene**	**968**	**968**	**1.94 ± 0.2**	**86**
**4**	**β-pinene ^a^**	**982**	**980**	**14.5 ± 1**	**94**
**5**	**Limonene ^a^**	**1030**	**1032**	**53 ± 5**	**93**
**6**	**γ-terpinene ^a^**	**1062**	**1064**	**5.9 ± 1**	**94**
7	terpinolene	1083	1085	0.2 ± 0.020	96
8	β-linalool^a^	1100	1101	0.2 ± 0.020	91
9	(E)-p-menth-2,8-dien-1-ol	1122	1123	0.13 ± 0.02	80
**10**	**limonene oxide, cis-**	**1130**	**1131**	**1 ± 0.3**	**96**
11	limonene oxide, trans-	1138	1138	0.7 ± 0.08	91
12	α-terpineol	1178	1179	0.3 ± 0.020	80
13	cis-carveol	1222	1222	0.3 ± 0.020	96
**14**	**citral ^a^**	**1240**	**1240**	**3.8 ± 0.9**	**96**
15	Δ-carvone	1242	1242	0.15 ± 0.01	93
16	nerol acetate	1363	1364	0.8 ± 0.05	91
17	geranyl aceate	1384	1385	0.9 ± 0.06	91
18	Caryophyllene ^a^	1415	1415	0.15 ± 0.01	99
19	α-bergamotene	1431	1430	0.21 ± 0.02	87
20	β-bisabolene ^a^	1504	1506	0.56 ± 0.04	95
21	caryophylleneoxyde	1596	1592	0.57 ± 0.05	91
	% Characterized	/	/	87.81	/
	Others	/	/	12.19	/

^a^ standard compound_._ Linear retention index (LRI) on HP-5MS column was experimentally determined using a homologous series of C10-C40 alkanes standard mixture (Van den Dool and Kratz, 1963). Arithmetic index (AI) was taken from Adams (2007) and/or the NIST 2017 Database (NIST 17, 2017. Mass Spectral Library (NIST/EPA/NIH). Gaithersburg, USA: National Institute of Standards and Technology. Last access 12_2021). Similarity index/mass spectrum (SI/MS) was compared with data reported in the NIST 2017 Database and were determined as previously reported [42,43]. Relative percentage values are means of three determinations with a structural equation modeling (SEM) in all cases below 10%.

## Data Availability

Data is contained within the article.

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
