# Peer review of "Virucidal Activity of Lemon Essential Oil against Feline Calicivirus Used as Surrogate for Norovirus"

_antibiotics, 2023, doi:10.3390/antibiotics12020322_

Round 1

Reviewer 1 Report

Nowadays, norovirus is global public health used for essential oils. The present study mainly focused on virucidal efficacy of lemon EO (LEO) against FCV was assessed in vitro. Totally 21 different components analyzed by GCMS. GCMS analysis results showed antioxidant, anticancer activities.

1.      Line number 21-23 need to improve the abstract section

2.      Line number 41-42 to change the sentences.

3.      183 line should change in detailed description?

4.      Norovirus is commonly affected in gastrointestinal not affected in anytime? Any reasons? Justify?

5.      What are the symptoms of norovirus infection, how long after exposure do they appear, and how long do they last? Justify?

6.      How norovirus disease is serious or pandemic? Add introduction section?

7.      What treatment is available for people with norovirus infection?

8.      Why did you selected GCMS analysis? Justification?

9.      The authors should add the GCMS figure in respective places.

10.  The relative percentage amount of each component was calculated by comparing its average peak area & retention time?

11.  What basis chosen for different concentrations of (30.20, 302.00 and 3020.00 μg/mL) Virucidal activity of Essential oils? Any reasons?

12.  Since essential oil is already being used as a food ingredient. What is the rationale behind choosing this for GCMS activity and cytotoxic activity?

13.  How are essential oils extracted from plants? Any other methods?

14.  What separation method is involved in the production of essential oil? Include in introduction section

15.  Why choosing various concentrations of the cytotoxicity of CrFK cells (8360, 4180, 2090, 1045, 522.50, 261.25, 130.63, 65.31, 32.66 μg/mL) at 72h? Any reasons?

16.  What is the mechanism behind the MTT?

17.  Conclusion should be revised for this work

Author Response

Reviewer 1

Nowadays, norovirus is global public health used for essential oils. The present study mainly focused on virucidal efficacy of lemon EO (LEO) against FCV was assessed in vitro. Totally 21 different components analyzed by GCMS. GCMS analysis results showed antioxidant, anticancer activities.

R1.1.      Line number 21-23 need to improve the abstract section

Reply to R1.1.  The sentence was improved (page 1 lines 22-23 revised manuscript).

R1.2.      Line number 41-42 to change the sentences.

Reply to R1.2.   This was done (page 1 line 44, page 2 line 45, revised manuscript).

R1.3.      line 183 should change in detailed description?

Reply to R1.3.  This was done (page 5 line 188, revised manuscript).

R1.4.      Norovirus is commonly affected in gastrointestinal not affected in anytime? Any reasons? Justify?

Reply to R1.4. Due to editorial constraints of the abstracts (word count limits) we could not provide more details and we had to be concise and generic. However, we provided more information in the text.

R1.5.      What are the symptoms of norovirus infection, how long after exposure do they appear, and how long do they last? Justify?

Reply to R1.5. The information was added in the text (page 2 lines 68-70, revised manuscript).

R1.6.      How norovirus disease is serious or pandemic? Add introduction section?

Reply to R1.6. A section describing the pandemic norovirus disease was added in the text (page 2 lines 70-72 revised manuscript).

R1.7.      What treatment is available for people with norovirus infection?

Reply to R1.7.  A section in the introduction regarding NoV treatment was added in the text (page 2 lines 83-87, revised manuscript).

R1.8.      Why did you select GCMS analysis? Justification?

Reply to R1.8. The composition of commercially available LEO used in our experiments has been confirmed by gas chromatography hyphenated with mass spectrometry (GC/MS) technique. The determination of essential oil chemotype (qualitative and quantitative characterization of EOs components) is a mandatory step for the quality assurance and standardization of pure essential oils in the production chain. Among all the chromatographic methods the conventional analytical methods based on GC/MS technique is the ultimate investigated method and it applies to the qualitative and quantitative classification of EOs based on the selectivity of their constituents. These methods are generally characterized by high performance in repeatability and accuracy as usually requested in this field (Waseem and Low, 2015 doi: 10.1002/jssc.201400724). This reference was numbered 41 in the revised manuscript.

R1.9. The authors should add the GCMS figure in respective places.

Reply to R1.9. The composition of LEO has been detailed in Table 1 in which the chromatographic parameters were reported. The chromatograms and printouts of essential oil used in this work are available for the Reviewers and Editors, but are strictly confidential. We are not authorized by the manufacturers to distribute them in this format.

R1.10.  The relative percentage amount of each component was calculated by comparing its average peak area & retention time?

Reply to R1.10. The qualitative and quantitative analyses of the LEO chemotype were carried out comparing the calculated Linear Retention Indices (LRIs) and Similarity Index of Mass Spectra (SI/MS) for the obtained peaks with the Arithmetic Index (AI) and the analogous data reported in literature and in NIST 2017 Databases (NIST 17, 2017. Mass Spectral Library - NIST/EPA/NIH. Gaithersburg, USA: National Institute of Standards and Technology. Last access 12_2022), respectively. Component relative percentages were calculated taking into account the area under the curve of each component GC peak to obtain a quantitative profile of LEO. The bibliography in the paragraph 2.3 Compound identification was revised.

R1.11.  What basis chosen for different concentrations of (30.20, 302.00 and 3020.00 μg/mL) Virucidal activity of Essential oils? Any reasons?

Reply to R1.11.  The maximum non-cytotoxic concentration was assessed and regarded as the concentration at which viability of the treated CrFK cells decreased to 20% with respect to the control cells (CC20). After exposing the cells to various concentrations of the compound (8360, 4180, 2090, 1045, 522.50, 261.25, 130.63, 65.31, 32.66 μg/mL) for 72 h, cytotoxicity was assessed by measuring the absorbance signal spectrophotometrically. Based on fitted dose–response curves, CC20 value of LEO was assessed at 30.20 μg/mL. The potential inhibitory effect of LEO against FCV was evaluated by pre-treatment of the virus with LEO at maximum non-cytotoxic dose (CC20) of 30.20 μg/mL and over the cytotoxic threshold (302.00 and 3020.00 μg/mL). The other two tested concentrations were 10-fold and 100-fold higher than CC20 since, if used as virucide, the compound is not to be posed into direct contact with the cells.

R1.12.  Since essential oil is already being used as a food ingredient. What is the rationale behind choosing this for GCMS activity and cytotoxic activity?

Reply to R1.12. The GC/MS technique was carried out for chemotype profiling of LEO as reported in our reply to R1.8. Cytotoxic activity has been evaluated as standard procedure for testing molecules on cells (Sharifi-Rad et al., 2017 doi: 10.14715/cmb/2017.63.8.10, Kubiça et al., 2014 doi: 10.1590/S1517-83822014005000030.). In this case it was done to discriminate also the confounding effects when assessing the residual infectivity of OE since we did not neutralize the activity of EO.

R1.13.  How are essential oils extracted from plants? Any other methods?

Reply to R1.13. This information was added in the introduction (page 1 lines 40-42, revised manuscript). The EOs could be obtained by different extraction methods ranging from distillation techniques to supercritical fluids or ultrasound and microwave-assisted extraction. Among all the extraction methods, hydrodistillation and stem distillation techniques remain the most used extraction methods to afford EOs for commercial and medicinal use. The European Pharmacopoeia has defined the EO as “odorant product, generally of a complex composition, obtained from a botanically defined plant raw material, either by driving by steam of water, either by dry distillation or by a suitable mechanical method without heating”. In particular, as regards the essential oil used in this study, it was obtained from stem distillation of peel.

R1.14.  What separation method is involved in the production of essential oil? Include in introduction section

Reply to R1.14.  See reply R1.13

R1.15.  Why choosing various concentrations of the cytotoxicity of CrFK cells (8360, 4180, 2090, 1045, 522.50, 261.25, 130.63, 65.31, 32.66 μg/mL) at 72 h? Any reasons?

Reply to R1.15.  The stock concentration of LEO was 8,360 μg/mL and we decided to use 2-fold dilutions of the compounds for the XTT assay that relies on at least five different concentrations of the compound to be able to evaluate maximum non-cytotoxic concentration (CC20) through non-linear curve fitting and dose-response curves.

R1.16.  What is the mechanism behind the MTT?

Reply to R1.16.  As described in the text (page 4 lines 154-156, revised manuscript) in this study we relied on the in vitro Toxicology Assay Kit (Sigma–Aldrich Srl, Milan, Italy), based on 3-(4,5-dimethylthiazol-2 yl)-2,5-diphenyl tetrazolium bromide (XTT). The XTT cell viability assay is based on 3- (4,5-dimethylthiazol-2 yl) -2,5-diphenyl tetrazolium bromide (XTT) which is metabolically reduced in viable cells in water-soluble formazan product, thus providing an indirect evidence of cell viability. In this assay, the number of viable cells is related to the intensity of color determined by photometric measurements of mitochondrial dehydrogenases.

R1.17.  Conclusion should be revised for this work

Reply to R1.17.  Conclusion have been implemented in text (page 9 lines 309-315, revised manuscript.

Author Response

Reviewer 2

Virucidal activity of lemon essential oil against feline calicivirus used as surrogate for Norovirus

R2.1 References 38 and 55 very old and need updating.

Reply to R2.1 References were replaced as requested.

Reference 38 was replaced by Waseem and Low, 2015 doi: 10.1002/jssc.201400724  renumbered as 41 in the revised manuscript.

Reference 55 was replaced by two references:

-Hofmann-Lehmann et al. 2022 doi: 10.3390/v14050937 (renumbered as 58 in the revised manuscript).

-Chiu et al., 2015 doi: 10.1016/j.ajic.2015.06.021 (renumbered 59 in the revised manuscript).

R2.2 The manuscript is devoid of conclusion at the end of the article.

Reply to R2.2 Conclusion have been implemented in text (page 9 lines 309-315, revised manuscript).

Reviewer 3 Report

1. Improve the introduction section. 

2. How have cultivatable caliciviruses been used as surrogates for in vitro evaluation of antivirals against Norovirus?

3. What was the chemical composition of lemon essential oil (LEO) found in this study?

4. What are the potential implications of these results for the development of alternative prophylaxis approaches for the control of Norovirus infection?

5. At what concentration and contact time did LEO show the most significant reduction in viral titre compared to the virus control?

6. Was there a statistically significant effect of LEO at different concentrations on viral titration of FCV?

Author Response

Reviewer 3

R3.1. Improve the introduction section. 

Reply to R3.1. The introduction has been improved following also the suggestions of R1 (see Reply to R1.2, R1.5, R1.6, R1.17).

R3.2. How have cultivatable caliciviruses been used as surrogates for in vitro evaluation of antivirals against Norovirus?

Reply to R3.2.  This information has been implemented in the text (page 2 line 91, page 3 lines 92-99, revised manuscript).   

R3.3. What was the chemical composition of lemon essential oil (LEO) found in this study?

Reply to R3.3. The composition of LEO has been detailed in Table 1 in which the chromatographic parameters were reported. The qualitative and quantitative analyses of the LEO chemotype were carried out comparing the calculated Linear Retention Indices (LRIs) and Similarity Index of Mass Spectra (SI/MS) for the obtained peaks with the Arithmetic Index (AI) and the analogous data reported in literature and in NIST 2017 Databases (NIST 17, 2017. Mass Spectral Library - NIST/EPA/NIH. Gaithersburg, USA: National Institute of Standards and Technology. Last access 12_2022), respectively. Component relative percentages were calculated taking into account the area under the curve of each component GC peak to obtain a quantitative profile of LEO. The bibliography in the paragraph 2.3 Compound identification was revised.

R3.4. What are the potential implications of these results for the development of alternative prophylaxis approaches for the control of Norovirus infection?

Reply to R3.4. The potential implications of these results are reported in the text (page 9 lines 313-315, revised manuscript).

R3.5. At what concentration and contact time did LEO show the most significant reduction in viral titre compared to the virus control?

Reply to R3.5. “LEO at 3020.0 μg/ml induced significant reductions of viral titre of 1.00 log10TCID50/50 μl (p<0.05) at 4h and of 1.25 log10TCID50/50 μl (p<0.05) at 8h when compared to virus control (6.75 log10 TCID50/50 μl) (Fig.1c)”. In detail, after 8h LEO at the highest concentration used in this study (3020.0 μg/ml) was able to decrease 5.3 106 viral particles/50 μl as compared to virus control.

R3.6. Was there a statistically significant effect of LEO at different concentrations on viral titration of FCV?

Reply to R3.6. As reported in the text “After 8 h, LEO at 302.00 μg/ml determined significant decrease of viral titre of 0.75 log10TCID50/50 μl (p < 0.05) with respect to virus control (6.75-7.00 log10 TCID50/50 μl) (Fig. 1b). LEO at 3020.0 μg/ml induced significant reductions of viral titre of 1.00 log10TCID50/50 μl (p<0.05) at 4h and of 1.25 log10TCID50/50 μl (p<0.05) at 8h when compared to virus control (6.75 log10 TCID50/50 μl) (Fig.1c).” LEO at lower concentration (30.20 μg/mL) at any contact time (10 min, 30 min, 1h, 4h, 8h) and at higher concentrations (302.00 and 3020.00 μg/ml) for other contact times not reported in the text did non induce significant reductions of viral titre (Figure 1).

Reviewer 4 Report

The manuscript entitled " Virucidal activity of lemon essential oil against feline calicivirus used as surrogate for Norovirus" and authored by Pellegrini and colleagues assessed the virucidal efficacy of the lemon essential oil (LEO) against feline calicivirus (FCV) in vitro.

In general, the topic of this paper is important. Although the results of this study may be promising, they are preliminary with limited investigations. Also, this paper suffers from major concerns.

Major concern

1-      Why the authors didn't validate their results with other investigations like electron microscope, qPCR,…….etc?

2-      The authors write “The chemical composition of LEO revealed the presence of 21 distinct molecules” which of these different molecules could be the main fraction for virucidal activity, against FCV? Why the authors don’t choose some of these active molecules to check their activity against FCV individually?

3-      The authors write “The pure EO of Citrus Lemon, also named as LEO, was provided by Specchiasol S.r.l. (Bussolengo, VR, Italy)” from which part of lemon (peel or lemon itself) pure EO is extracted?

4-      The authors write “The results in this study are difficult to be compared with those from other reports due to the different conditions used in the experiments i.e., temperatures, the number of viral particles, the EO/virus contact times. Conservation of EOs is a limiting factor as they are very sensitive to heat, light, oxygen and humidity and are characterized by a remarkable rapidity of evaporation”. How did the authors confirm that all these variable condition were controlled in their study?

5-      I recommend other experiments should be performed to validate the results due to variable chemical composition of LEO and conditions of extraction of LEO.

Minor concern:

6-      Please write in brief about the possible mechanism of action of virucidal activity of lemon essential oil in discussion part.

7-      Please write the title of table 1 above the table, not below it.

8-      Please add the abbreviation of FCV and CrFK in figure legend 1.

Author Response

Reviewer 4

The manuscript entitled " Virucidal activity of lemon essential oil against feline calicivirus used as surrogate for Norovirus" and authored by Pellegrini and colleagues assessed the virucidal efficacy of the lemon essential oil (LEO) against feline calicivirus (FCV) in vitro.

In general, the topic of this paper is important. Although the results of this study may be promising, they are preliminary with limited investigations. Also, this paper suffers from major concerns.

Major concern

R4.1.      Why the authors didn't validate their results with other investigations like electron microscope, qPCR, …, …. etc?

Reply to R4.1.  We agree with the referee that qPCR is a good proxy to assess the extent of replication of the virus and, in several cases, we couple molecular quantification (qPCR) with virus titration. However, since qPCR cannot distinguish between infectious and noninfectious viral particles, virus titration in cells is the elective option to assess residual infectivity. In the case of non-cultivatable viruses, surrogate strategies can be implemented, including FCV that is easily cultivatable in CrFK cells.   

R4.2.      The authors write “The chemical composition of LEO revealed the presence of 21 distinct molecules” which of these different molecules could be the main fraction for virucidal activity, against FCV? Why the authors don’t choose some of these active molecules to check their activity against FCV individually?

Reply to R4.2. We agree with the referee’ comment. In the discussion (page 8 line 298-307) we have mentioned the possibility that the antiviral activity of LEO may depend on a few fractions and that this could be worth investigating. On the other hand, based on the literature we know that EOs are a complex mixture of small molecules, whose synthesis occurs during the secondary metabolism pathway by secretory cells of aromatic plants. In the mixture, such compounds are present at different concentrations. The primary active component of LEO was limonene that accounted for more than 50% of the 21 components in our study. Although the chemical structure of the single EO components affects their precise mode of action and biological activity, it is most likely that EOs biological activity strictly depends on the phytocomplex, and is the result of separate and different mechanisms determined by synergic interactions with different cell targets.

R4.3.      The authors write “The pure EO of Citrus Lemon, also named as LEO, was provided by Specchiasol S.r.l. (Bussolengo, VR, Italy)” from which part of lemon (peel or lemon itself) pure EO is extracted?

Reply to R4.3. As regards the essential oil used in this study, as claimed in the package leaflet, it was obtained from stem distillation of Citrus Lemon peel. We added this in the materials and methods Heading 2.1. “Analysis of LEO”.

R4.4. The authors write “The results in this study are difficult to be compared with those from other reports due to the different conditions used in the experiments i.e., temperatures, the number of viral particles, the EO/virus contact times. Conservation of EOs is a limiting factor as they are very sensitive to heat, light, oxygen and humidity and are characterized by a remarkable rapidity of evaporation”. How did the authors confirm that all these variable conditions were controlled in their study?

Reply to R4.4. This is a limit of these studies and we disclosed this in the discussion (page 7 lines 268-274, revised manuscript) as commented by the referee.

R4.5. I recommend other experiments should be performed to validate the results due to variable chemical composition of LEO and conditions of extraction of LEO.

Reply to R4.5.

The EOs could be obtained by different extraction methods ranging from distillation techniques to supercritical fluids or ultrasound and microwave-assisted extraction. Among all the extraction methods, hydrodistillation and stem distillation techniques remain the most used extraction methods to afford EOs for commercial and medicinal use. According to environmental and plant living conditions, climate, soil, cultivation techniques and to the level of expertise and care given by farmers and distillers during the extraction steps, the same plant species may show significant chemical differences in their qualitative and quantitative EO composition, which is defined as EO chemotype. In particular, the essential oil used in our experiments is commercially available and it is obtained by stem distillation of peel according with the European Pharmacopoeia definition see reply R4.3. Before their use we check and report the composition of essential oil using GC/MS technique. We have disclosed this limit in the discussion (page 7 lines 268-274, revised manuscript).

Minor concern

R4.6.      Please write in brief about the possible mechanism of action of virucidal activity of lemon essential oil in discussion part.

Reply to R4.6. In order to explain the possible mechanism of action of virucidal activity of LEO we added a short paragraph in the text (page 7 line 280, page 8 lines 281-283, revised manuscript)

R4.7.      Please write the title of table 1 above the table, not below it.

Reply to R4.7.  This was done.   

R4.8.      Please add the abbreviation of FCV and CrFK in figure legend 1.

Reply to R4.8. This was done.   

Round 2

Reviewer 3 Report

Accept in its current form.

Reviewer 4 Report

The manuscript entitled " Virucidal activity of lemon essential oil against feline calicivirus used as surrogate for Norovirus" and authored by Pellegrini and colleagues assessed the virucidal efficacy of the lemon essential oil (LEO) against feline calicivirus (FCV) in vitro.

The revised manuscript is improved compared to prior revision. My comments were adequately answered by the authors. Therefore, I consider that the revised manuscript is acceptable and suitable for publication in Antibiotics Journal.